# Validation of a Polish version of the National Institutes of Health Stroke Scale: Do moderate psychometric properties affect its clinical utility?

**Adam Wiśniewski**[1]*, **Karolina Filipska**[2], **Marlena Puchowska**[1], **Katarzyna Piec**[1], **Filip Jaskólski**[1], **Robert Ślusarz**[2]

**1** Department of Neurology, Laboratory for Experimental Biotechnology, Collegium Medicum in Bydgoszcz, Nicolaus Copernicus University in Toruń, Bydgoszcz, Poland, **2** Department of Neurological and Neurosurgical Nursing, Laboratory for Experimental Biotechnology, Collegium Medicum in Bydgoszcz, Nicolaus Copernicus University in Toruń, Bydgoszcz, Poland

* adam.lek@wp.pl

**Data Availability Statement:** All data files are available from Zenodo (DOI: 10.5281/zenodo. 4536002).

## Abstract

### Background

The National Institutes of Health Stroke Scale (NIHSS) is a validated tool for assessing the severity of stroke. It has been adapted into several languages; however, a Polish version with large-scale psychometric validation, including repeatability and separate assessments of anterior and posterior stroke, has not been developed. We aimed to adapt and validate a Polish version of the NIHSS (PL-NIHSS) while focusing on the psychometric properties and site of stroke.

### Methods

The study included 225 patients with ischemic stroke (102 anterior and 123 posterior circulation stroke). Four NIHSS-certified researchers estimated stroke severity using the most appropriate scales to assess the psychometric properties (including internal consistency, homogeneity, scalability, and discriminatory power of individual items) and ultimately determine the reliability, repeatability, and validity of the PL-NIHSS.

### Results

The PL-NIHSS achieved Cronbach's alpha coefficient of 0.6885, which indicates moderate internal consistency and homogeneity. Slightly more than half of the individual items provided sufficient discriminatory power (r > 0.3). A favorable coefficient of repeatability (0.6267; 95% confidence interval: 0.5737–0.6904), narrow limits of inter-rater agreement, and excellent intraclass correlation coefficients or weighted kappa values (> 0.90), demonstrated high reliability of PL-NIHSS. Highly significant correlations with other tools confirmed the validity and predictive value of the PL-NIHSS. In posterior stroke, the PL-NIHSS achieved the required Cronbach's alpha coefficient (0.71070). Additionally, stroke location did not affect other psychometric features or instrument reliability and validity.

**Funding:** The author(s) received no specific funding for this work.

**Competing interests:** The authors have declared that no competing interests exist.

## Conclusions

We developed a valid and reliable tool for assessing stroke severity in Polish-speaking participants. Moderate psychometric features were emphasized without limiting its clinical applications.

## Introduction

Clinometric scales are used to objectively evaluate the severity of stroke. Undoubtedly, the National Institutes of Health Stroke Scale (NIHSS) has played the most important role in stroke assessment for several years [1]. It is widely accepted due to its simplicity, high reproducibility, and ease of performance [2] and was designed to be used not only by neurologists, but also by other thoroughly trained members of the stroke team [3]. Furthermore, apart from delivering an objective and reliable estimation of stroke severity, numerous studies have stressed the usefulness of the NIHSS in assessing the clinical prognoses, outcomes, and risks for large intracranial vessel occlusions, thus, emphasizing its predictive value [4, 5]. Several researchers from various countries have adapted and validated the NIHSS after demonstrating its high reproducibility and highlighting its clinical utility [6–15]. However, less attention has been focused on the psychometric properties, such as internal consistency or the discriminatory power of individual items, because these factors have only been analyzed in individual reports [16]. Notably, the NIHSS psychometric parameters that determine homogeneity, stability, and individual component discriminatory power are equally important as its overall utility and clinical validity. Obtaining the appropriate values for all these components will determine the overall quality of the diagnostic tool, and it is of utmost importance that these features are independent of the language, country, region, and culture. In light of this observation, the lack of a reliable and in-depth analysis of the NIHSS scale is a shortcoming; therefore, a comprehensive assessment of the NIHSS is essential to better define its structural features as well as overall clinical and practical relevance.

The language barrier and lack of a standardized stroke evaluation tool in Poland have resulted in a clinical need for a reliable and valid instrument that can enable members of the stroke team in evaluating Polish-speaking patients. The aim of the current study was to develop and validate a Polish version of the NIHSS (PL-NIHSS) and to assess its psychometric properties, including internal consistency, homogeneity, and scalability in relation to its overall reliability and clinical accuracy.

## Methods

### Study design and participants

This prospective, observational, single-center study was conducted between December 2019 and August 2020 in the Stroke Unit of the Department of Neurology at the University Hospital No. 1, Bydgoszcz, Poland. We enrolled 225 patients with ischemic stroke, including 102 patients with anterior and 123 patients with posterior circulation stroke. All participants met the requirements of the updated definition of stroke proposed by the American Heart and Stroke Association [17].

The clinical and functional parameters were assessed within 24 hours of stroke onset using the PL-NIHSS and Glasgow Coma Scale (GCS). The questionnaires were completed by four investigators, including two stroke physicians, a stroke research nurse, and a physiotherapist,

all of whom were NIHSS-certified and had several years of experience in the intensive stroke unit.

Estimation of the inter-rater reliability of the PL-NIHSS was based on evaluations by three randomly selected researchers. The time difference between each assessment did not exceed 2 hours. Repeatability was assessed by analyzing the total PL-NIHSS values assessed by two randomly selected examiners. Three hours later, one researcher randomly selected from the initial three researchers re-assessed the patient (test-retest) using the PL-NIHSS to estimate intra-rater reliability. Subsequently, a randomly selected researcher (from the total group of researchers) evaluated the patient within the first 24 hours of onset of stroke using the GCS to evaluate its construct validity and again at 3 months using the Barthel Index and modified Rankin Scale (mRS) to assess its predictive validity.

The following exclusion criteria were used: (1) significant speech impairment or disturbances of consciousness that prevented a patient from providing informed consent to participate in the study, and (2) patients undergoing specific stroke therapy (intravenous thrombolysis and/or endovascular treatment), which can significantly contribute to discernable fluctuations in the clinical condition. The baseline characteristics of the participants are summarized in Table 1.

## PL-NIHSS

Adaptation of the English version of the NIHSS into Polish was performed in accordance with standards proposed by the International Quality of Life Assessment Project [18]. Two forward translations were used to create an intermediate version that was translated back for comparison

**Table 1. Baseline characteristics of ischemic stroke subjects (n = 225).**

| Parameter | Value |
|---|---|
| Age, median (range) | 70 (34–97) |
| Sex: | |
| Male, N (%) | 120 (53.3%) |
| Female, N(%) | 105 (46.7%) |
| Stroke etiology: | |
| Large vessels disease, N (%) | 31 (13.8%) |
| Small vessel disease, N(%) | 79 (35.1%) |
| Cardioembolism, N(%) | 66 (29.3%) |
| Other, N(%) | 49 (21.8%) |
| NIHSS on admission, median (range) | 4 (1–21) |
| mRS on admission, median (range) | 2 (0–5) |
| Barthel Index od admission, median (range) | 85 (5–100) |
| Risk factors: | |
| Hypertension, N(%) | 179 (79.5%) |
| Diabetes, N(%) | 74 (32.9%) |
| Smoking, N(%) | 61 (27.1%) |
| Ischemic heart disease, N(%) | 40 (17.8%) |
| Hyperlipidemia, N(%) | 62 (28.9%) |
| Obesity, N(%) | 54 (24.0%) |
| BMI, median (range) | 27.25 (19.22–39.64) |
| Alcohol abuse, N(%) | 21 (9.3%) |
| Atrial fibrillation, N(%) | 66 (29.3%) |

NIHSS- the National Institutes of Health Stroke Scale; mRS- modified Rankin Scale; BMI- Body Mass Index.

with the original version. After analyzing for any contradictions or misinterpretations and obtaining agreement on the consistency and equivalence, the scale was reviewed by Polish-speaking neurologists who estimated how well it was comprehended and rated its overall acceptance. Each item received the required minimum of three points (out of a total of four points) in the content validity rating [19], and after considering minor corrections and suggestions, a preliminary version of the PL-NIHSS was established (S1 Table). Subsequently, the items that assessed speech disorders, inattention, or visual extinction (Fig 1) were modified and adapted to the cultural aspects that would be better recognized and understood by the Polish population. The word complexity, knowledge of phrases, and commonness of idioms were considered while maintaining the content and meaning of the original items. The researchers completed the PL-NIHSS training based on repeated clinical examinations of all the items. The same rules were also adapted for the assessment of individual components included in the original NIHSS [20].

## Ethical statement

The study protocol was approved by the Bioethics Committee of the Nicolaus Copernicus University in Torun at Collegium Medicum of Ludwik Rydygier in Bydgoszcz (KB number 732/2019). All participants read and understood the study protocol and provided informed written consent to participate in the study.

## Statistical evaluation methods

STATISTICA v13.1 (Dell Technologies, Round Rock, TX, USA) was used for the statistical analyses. The following tests were performed: Spearman's rank correlation (estimation of construct and predictive validity), intraclass correlation coefficient (evaluation of inter-rater and intra-rater agreement), and weighted Cohen's kappa (intra-rater agreement). Cronbach's alpha coefficient and Bland–Altman analysis were performed to assess the psychometric properties of the PL-NIHSS [21, 22]. A p-level $< 0.05$ was considered statistically significant.

## Results

A Cronbach's alpha coefficient of 0.6885 was achieved in all patients with stroke with individual values of 0.6387 and 0.7107 for anterior and posterior stroke, respectively. The characteristics of individual items are summarized in Table 2.

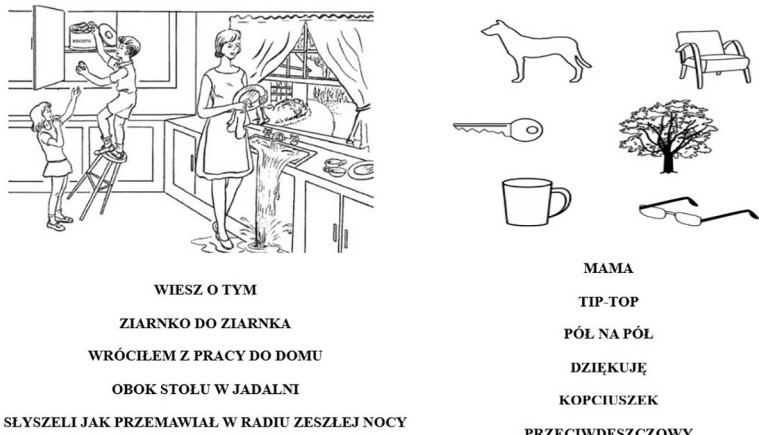

**Fig 1. Pictogram showing modified words, phrases, and pictures for better assessment of speech disorders, inattention, and extinction in a Polish-speaking population.**

**Table 2. Psychometric properties of individual items of the Polish version of the National Institutes of Health Stroke Scale (PL-NIHSS).**

| Item | Discriminant power of item | Cronbach's alpha when item is removed | Discriminant power of item | Cronbach's alpha when item is removed | Discriminant power of item | Cronbach's alpha when item is removed |
|---|---|---|---|---|---|---|
| | Total | Total | Anterior stroke | Anterior stroke | Posterior stroke | Posterior stroke |
| | (n = 225) | (n = 225) | (n = 102) | (n = 102) | (n = 123) | (n = 123) |
| LOC | 0.294136 | 0.683330 | 0.341463 | 0.629911 | 0.296433 | 0.704423 |
| LOC questions | 0.265276 | 0.680356 | 0.207420 | 0.630403 | 0.192353 | 0.708738 |
| LOC commands | 0.328857 | 0.679176 | 0.311585 | 0.624007 | 0.194121 | 0.710916 |
| Best gaze | 0.477950 | 0.653909 | 0.670050 | 0.568623 | 0.317488 | 0.695873 |
| Visual | 0.255057 | 0.679979 | 0.664460 | 0.560121 | 0.089200 | 0.751243 |
| Facial palsy | 0.663285 | 0.631123 | 0.492282 | 0.615787 | 0.784627 | 0.629697 |
| Motor arm of left | 0.464550 | 0.647010 | 0.279896 | 0.631653 | 0.579039 | 0.655805 |
| Motor arm of right | 0.181131 | 0.693888 | -0.039626 | 0.692250 | 0.307756 | 0.696919 |
| Motor leg of left | 0.473464 | 0.644074 | 0.280524 | 0.626452 | 0.607063 | 0.655786 |
| Motor leg of right | 0.220941 | 0.684049 | 0.026909 | 0.668041 | 0.349657 | 0.693387 |
| Limb ataxia | -0.374303 | 0.753991 | -0.062611 | 0.644287 | 0.012107 | 0.743963 |
| Sensory | 0.397524 | 0.667285 | 0.421887 | 0.601444 | -0.063696 | 0.716220 |
| Best language | 0.204271 | 0.683126 | 0.093520 | 0.644686 | 0.207292 | 0.708681 |
| Dysarthria | 0.597656 | 0.632055 | 0.438402 | 0.602290 | 0.673961 | 0.636620 |
| Extinction and inattention | 0.570273 | 0.650585 | 0.679698 | 0.571650 | 0.385419 | 0.692340 |

LOC- level of consciousness.

In the group that included both types of stroke (irrespective of location), only 8/15 (53.3%) items achieved a satisfactory and required discriminant level (r>0.3) [23]. Of those, only three, including items for facial palsy, dysarthria, and extinction or inattention, achieved a high correlation with the others (r>0.5). Limb ataxia was the least correlated with the other components. However, when limb ataxia and right arm motor function were excluded, the overall alpha coefficient increased. In the patients with anterior stroke, eight items met the minimum requirements for discriminatory power; of those, only items for visual field, best gaze, and extinction or inattention achieved high values. Notably, the motor function of the right arm and limb ataxia were distinguished from the other items by negative correlation values. Removing four items (motor function of right arm, motor function of right leg, limb ataxia, and best language) improved the overall accuracy of the PL-NIHSS. In the patients with posterior stroke, eight items achieved a satisfactory discriminant level, and half of them, including items for facial palsy, motor function of the left arm, motor function of the left leg, and dysarthria were highly correlated with the others. Only one item (sensory) was negatively correlated with the others; however, removing four items (sensory, limb ataxia, level of consciousness-commands, and visual field) increased the overall alpha coefficient. The median inter-item correlation for the entire stroke group was 0.1834, while the values were 0.1807 and 0.1737 for anterior and posterior stroke, respectively.

The results of the inter-rater and intra-rater agreements are summarized in Table 3.

Excellent weighted kappa values ($\kappa > 0.9$) and intraclass correlation coefficients (ICC $> 0.9$) among all the items indicated high reproducibility of the PL-NIHSS. A favorable coefficient of repeatability (CR = 0.6267; 95% confidence interval [CI] = 0.5737–0.6904) and narrow limits of agreement (lower: -0.6408, 95%CI = -0.7128 to -0.5689; upper: 0.6142, 95% CI = 0.5422–0.6862) were observed in Bland–Altman analyses (Fig 2), thus, emphasizing the

**Table 3. Inter-rater and intra-rater reliability of the Polish version of the National Institutes of Health Stroke Scale (PL-NIHSS).**

| ITEM | INTER-RATER | | INTRA-RATER | | | |
|---|---|---|---|---|---|---|
| | RELIABILITY | | RELIABILITY | | | |
| | ICC | 95% CI | ICC | 95% CI | weighted κ | 95% CI |
| LOC | 1.00 | - | 1.00 | - | 1.00 | - |
| LOC questions | 0.9854 | 0.9818–0.9884 | 0.9780 | 0.9715–0.9830 | 0.9708 | 0.9137–1.000 |
| LOC commands | 0.9813 | 0.9767–0.9851 | 0.9639 | 0.9534–0.9721 | 0.9543 | 0.8652–1.000 |
| Best gaze | 0.9897 | 0.9872–0.9918 | 0.9852 | 0.9808–0.9886 | 0.9493 | 0.8812–1.000 |
| Visual | 0.9925 | 0.9906–0.9940 | 0.9887 | 0.9853–0.9913 | 0.9574 | 0.9009–1.000 |
| Facial palsy | 0.9772 | 0.9717–0.9819 | 0.9789 | 0.9727–0.9837 | 0.9745 | 0.9465–1.000 |
| Motor arm of left | 0.9935 | 0.9919–0.9948 | 0.9941 | 0.9924–0.9955 | 0.9745 | 0.9469–1.000 |
| Motor arm of right | 0.9917 | 0.9897–0.9934 | 0.9782 | 0.9718–0.9832 | 0.9528 | 0.9184–1.000 |
| Motor leg of left | 0.9937 | 0.9922–0.9950 | 0.9907 | 0.9879–0.9928 | 0.9636 | 0.9293–1.000 |
| Motor leg of right | 0.9916 | 0.9895–0.9933 | 0.9824 | 0.9772–0.9865 | 0.9516 | 0.9059–1.000 |
| Limb ataxia | 0.9875 | 0.9844–0.9901 | 0.9816 | 0.9762–0.9858 | 0.9688 | 0.9387–1.000 |
| Sensory | 0.9918 | 0.9898–0.9935 | 0.9876 | 0.9839–0.9904 | 0.9626 | 0.8915–1.000 |
| Best language | 0.9876 | 0.9846–0.9902 | 0.9901 | 0.9871–0.9923 | 0.9754 | 0.9285–1.000 |
| Dysarthria | 0.9902 | 0.9877–0.9922 | 0.9852 | 0.9808–0.9888 | 0.9774 | 0.9519–1.000 |
| Extinction and inattention | 0.9854 | 0.9818–0.9984 | 0.9778 | 0.9712–0.9828 | 0.9357 | 0.8495–1.000 |

LOC- level of consciousness; ICC- intraclass correlation coefficient

CI- Confidence Interval; κ- Cohen's kappa value.

accuracy of PL-NIHSS. A vast majority of related pairs of total scores (n = 211; 93.8%) fell within the limits of agreement and reached an identical total number of points whereas the maximum difference in the total score between the examiners was two points, which was observed only in three cases.

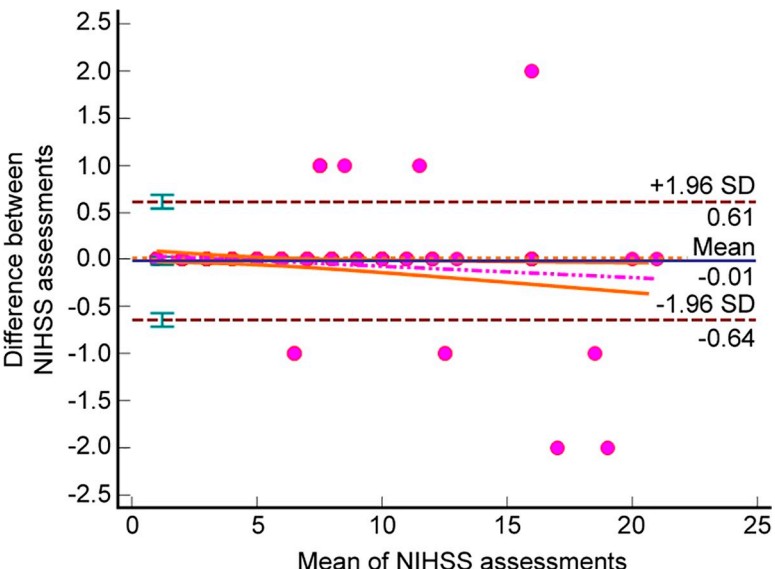

**Fig 2. Bland–Altman diagram indicating the repeatability of the Polish version of the National Institutes of Health Stroke Scale (PL-NIHSS).** The distribution of plots is based on the mean and difference from the total PL-NIHSS scores obtained by two randomly selected examiners. The limits of agreement occupy the area between the dashed lines. The 95% confidence interval of the regression line is located between the orange bold lines.

We observed a moderate, but significant correlation between the PL-NIHSS score and the initial GCS score (r = -0.4460, p < 0.0001), which indicated satisfactory construct validity (Fig 3A). On the 90th day after the onset of stroke, we also observed a high correlation between the PL-NIHSS, Barthel Index (r = -0.8648, p < 0.0001), and mRS (r = 0.8310, p < 0.0001), which reflected the predictive validity of the device (Fig 3B and 3C). We found no significant differences in the assessment of the reliability (ICC, kappa, CR, limits of agreement) or validity (correlation coefficient) between the patient groups with anterior and posterior stroke as well as in comparison of each subgroup with the overall group.

## Discussion

To our knowledge, this study describes the first adaptation and validation of a Polish version of the NIHSS (PL-NIHSS). In this novel report, we highlighted its moderate psychometric properties, assessed its repeatability using Bland–Altman statistics, and analyzed its internal consistency, reliability, and validity based on the stroke location (anterior or posterior).

An ideally constructed stroke scale should be characterized by appropriate psychometric parameters, which demonstrate the correct structure of the tool. Particularly, it should be characterized by scalability (internal consistency and homogeneity) by confirming that each component of the instrument is equally important and measures the same attribute [24]. According to Nunnally's principle, the Cronbach's alpha coefficient used for this assessment should reach a minimum of 0.7 [21]. Each item on the scale should also significantly correlate with the others (discriminatory power), and its removal should not increase the overall reliability of the scale. We observed a sufficient alpha coefficient only in posterior stroke (slightly exceeding the limit), whereas the required value was not achieved in the groups with anterior and overall stroke. Only slightly more than half of the assessed items had appropriate discriminatory power in the overall stroke group as well as in the anterior and posterior stroke subgroups. Additionally, some items did not correlate with the others at all, thus, contributing to a reduction in the quality of the entire tool. The median correlation coefficients were far below those expected. Our findings emphasized doubtful homogeneity of the adapted version of the NIHSS and are inconsistent with the data reported by Sun et al. [16] who demonstrated Cronbach's alpha coefficient of 0.92 and mean inter-item correlation of 0.44. However, they analyzed only 48 patients with stroke, and the small sample size may have significantly affected their overall study reliability [25]. The moderate psychometric properties observed in our study indicated a lack of homogeneity and internal consistency, and therefore, suggests a structural disadvantage of the NIHSS. Accordingly, further research to improve the existing NIHSS version should be supported in order to develop a scalable tool in accordance with the current international guidelines.

Irrespective of the design imperfections, the significant clinical utility of the validated version of the NIHSS should be emphasized; it was particularly manifested in the high reliability and validity observed in our study. Our findings are consistent with those of other studies in this topic; however, we noted higher individual item agreement values than those reported by most other investigators. Only one report by Jurjans et al. [26] found that all the items of a Latvian validated tool achieved excellent ICC (> 0.95) in both inter-rater and intra-rater assessments. The authors of validation reports of other scales found moderate, and sometimes, even poor agreement between the selected items [9–15]. Notably, the sample size in the present and the Latvian study were larger than those in the other studies, thus, emphasizing the significance of the results in this study as well as highlighting the high reproducibility of the PL-NIHSS. Simultaneously, our research supports the wide use and assessment of the NIHSS by qualified, trained, and certified members of the stroke team and not just neurologists. A

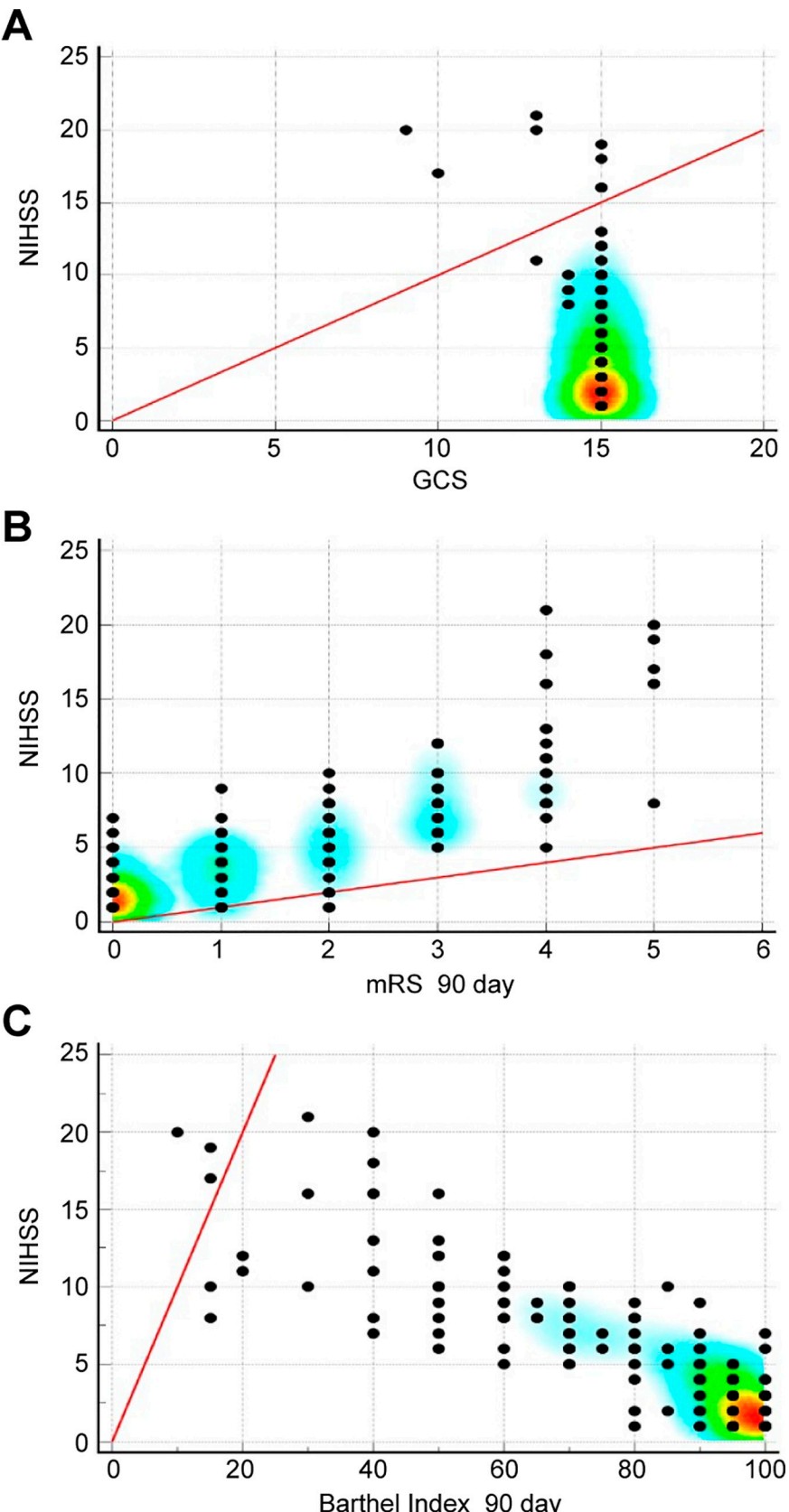

**Fig 3.** Construct (A) and predictive (B, C) validity of the Polish version of the National Institutes of Health Stroke Scale (PL-NIHSS). Significant correlation with Glasgow Coma Scale (GCS) on the first day of stroke. Significant correlations with (B) modified Rankin Scale (mRS) and (C) Barthel index on the 90th day of stroke.

clear advantage of our study over others is the assessment of repeatability based on the agreement achieved between raters regarding the total score and not just individual items. To our knowledge, this is the first study to emphasize a satisfactory coefficient of repeatability and narrow limits of agreement using Bland–Altman statistics, thus, confirming the stability and reliability of the validated tool. The high construct and predictive validity of the PL-NIHSS was reflected in the significant, high, and moderate correlations with other instruments used in similar situations in other studies.

Another strength of our study is the assessment of the psychometric parameters, reliability, and validity depending on the stroke location. Many reports have demonstrated that the NIHSS is more accurate when used to assess the severity of anterior stroke whereas the clinical condition of posterior stroke is often underestimated [27]. Therefore, unlike previous studies, we attempted to validate the PL-NIHSS with both types of stroke and found that specifying the type of stroke did not negatively affect the parameters, thus, confirming the reproducibility, repeatability, and validity of the tool. This result verified the high accuracy of the validated instrument, irrespective of the area of brain vascularization. Surprisingly, better psychometric properties, such as internal consistency or homogeneity, were noted in the patients with posterior stroke. These differences between the compared groups confirmed that better scalability of the PL-NIHSS did not translate into a more accurate assessment of stroke severity or increase its validity and reliability. Furthermore, we hypothesized that the psychometric properties of the validated instrument did not affect or limit its clinical utility. Nevertheless, we believe that the optimal situation occurs when the commonly used scale is characterized by high psychometric values as well as high reliability and validity.

The current study has some limitations. The study sample size was moderate, although it was larger than in those in other studies. Our study was a single-center study; therefore, verification of our postulates, particularly regarding the psychometric aspects, is required in multi-center studies, preferably with international cooperation. Due to the requirement for obtaining informed written consent, some patients with stroke were procedurally excluded, and therefore, the data did not cover the entire stroke profile (especially of patients with severe strokes).

## Conclusions

We developed a valid and reliable Polish version of the NIHSS suitable for use in everyday practice by trained and certified staff of the Polish-speaking stroke unit. The moderate psychometric properties emphasized in the PL-NIHSS did not affect its clinical usefulness. However, considering the international requirements for commonly used diagnostic tools, further research should be pursued to improve the design and structural quality of the current NIHSS.

## Supporting information

**S1 Table. Polish version of the National Institutes of Health Stroke Scale.**
(PDF)

## Acknowledgments

Special thanks to the Members of the Student Research Club at the Department of Neurology at Collegium Medicum in Bydgoszcz for contributing to the development of the database.

## Author Contributions

**Conceptualization:** Adam Wiśniewski.

**Data curation:** Adam Wiśniewski, Marlena Puchowska, Katarzyna Piec, Filip Jaskólski.

**Formal analysis:** Adam Wiśniewski, Karolina Filipska, Filip Jaskólski.

**Investigation:** Adam Wiśniewski, Karolina Filipska, Katarzyna Piec.

**Methodology:** Adam Wiśniewski, Marlena Puchowska, Filip Jaskólski.

**Project administration:** Adam Wiśniewski.

**Resources:** Adam Wiśniewski, Katarzyna Piec.

**Supervision:** Adam Wiśniewski, Robert Ślusarz.

**Validation:** Adam Wiśniewski.

**Visualization:** Adam Wiśniewski, Robert Ślusarz.

**Writing – original draft:** Adam Wiśniewski.

**Writing – review & editing:** Adam Wiśniewski, Karolina Filipska, Robert Ślusarz.

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
