## [Decision Letter · Decision Letter 0]

10 Feb 2021

PONE-D-21-00869

Validation of a Polish version of the National Institutes of Health Stroke Scale- can moderate psychometric properties affect its clinical utility?

PLOS ONE

Dear Dr. Wiśniewski,

Thank you for submitting your manuscript to PLOS ONE. After careful consideration, we feel that it has merit but does not fully meet PLOS ONE’s publication criteria as it currently stands. Therefore, we invite you to submit a revised version of the manuscript that addresses the points raised during the review process.

The manuscript seems interesting, but I encourage the authors to seek for English Editing Services to correct grammatical errors through out the manuscript. Also, the quality of the images is low, and make them difficult to evaluate, please refer to PLOS ONE guidelines for images quality.

We look forward to receiving your revised manuscript.

Kind regards,

Miguel A. Barboza, MD, MSc

Academic Editor

PLOS ONE

Journal Requirements:

Reviewers' comments:

Reviewer's Responses to Questions

**Comments to the Author**

1. Is the manuscript technically sound, and do the data support the conclusions?

Reviewer #1: Yes

2. Has the statistical analysis been performed appropriately and rigorously? 

Reviewer #1: Yes

3. Have the authors made all data underlying the findings in their manuscript fully available?

Reviewer #1: Yes

4. Is the manuscript presented in an intelligible fashion and written in standard English?

Reviewer #1: Yes

5. Review Comments to the Author

Reviewer #1: The paper is interesting, well written and valuable. Results sound not too optimistic not too pessimistic, so PL-NIHSS may be used until new version of the scale will be prepared. It is first time done for Polish version.

I have one important comment.

Authors are at day of admission, beside routine patients’ characteristic and PL- NIHSS are also doing mRS, Barthel and GCS. Writing in page 4, in methods. „which are most widely accepted tools for this purpose” not giving references. GCS may be done, but mRS and Barthel cannot be performed in most cases and is unreliable. At admission we ask patient or family and notice – pre stroke functional status.

I propose to delate these scales from admission characteristic, GCS can stay (not necessary), as well from results and figures. Is Ok to do mRS and Barthel after 90 days, but authors do not mention it in the methods, but we see in results.

6. PLOS authors have the option to publish the peer review history of their article (what does this mean?). If published, this will include your full peer review and any attached files.

Reviewer #1: No

---

## [Author Response · Author response to Decision Letter 0]

19 Feb 2021

Response to Reviewers comments

Academic Editor:

The manuscript seems interesting, but I encourage the authors to seek for English Editing Services to correct grammatical errors through out the manuscript. Also, the quality of the images is low, and make them difficult to evaluate, please refer to PLOS ONE guidelines for images quality.

Response:

Dear Editor,

Thank You for this comment. According to the Editor recommendation our manuscript underwent the extensive language, style, grammar and typographical corrections made by Editage- Professional English Editing Service. We have also improved the quality and resolution of the Figures and checked them with PACE tool to ensure that they meet PLOS requirements. We also provided the changes suggested by the Reviewer, regarding stroke assessment at admission.

Reviewer:

The paper is interesting, well written and valuable. Results sound not too optimistic not too pessimistic, so PL-NIHSS may be used until new version of the scale will be prepared. It is first time done for Polish version. I have one important comment.

Authors are at day of admission, beside routine patients’ characteristic and PL- NIHSS are also doing mRS, Barthel and GCS. Writing in page 4, in methods. „which are most widely accepted tools for this purpose” not giving references. GCS may be done, but mRS and Barthel cannot be performed in most cases and is unreliable. At admission we ask patient or family and notice – pre stroke functional status.

I propose to delate these scales from admission characteristic, GCS can stay (not necessary), as well from results and figures. Is Ok to do mRS and Barthel after 90 days, but authors do not mention it in the methods, but we see in results.

Response:

Thank You for positive opinion regarding our paper.

As suggested by the Reviewer we have removed Barthel Index and modified Rankin Scale from admission characteristics, both in the Methodology Section, as well as in the Results Section and Figures. At this moment there are only PL-NIHSS and Glasgow Coma Scale as stroke assessment at admission. The stroke evaluation after 90 days is described in Methodology Section- Lines- 84-87.

I would like to thank You for the careful review of our study and the constructive comments that have been used to organize all issues and improve the work.

---

## [Decision Letter · Decision Letter 1]

15 Mar 2021

Validation of a Polish version of the National Institutes of Health Stroke Scale: Do moderate psychometric properties affect its clinical utility?

PONE-D-21-00869R1

Dear Dr. Wiśniewski,

We’re pleased to inform you that your manuscript has been judged scientifically suitable for publication and will be formally accepted for publication once it meets all outstanding technical requirements.

Kind regards,

Miguel A. Barboza, MD, MSc

Academic Editor

PLOS ONE

Additional Editor Comments (optional):

Reviewers' comments:

Reviewer's Responses to Questions

**Comments to the Author**

1. If the authors have adequately addressed your comments raised in a previous round of review and you feel that this manuscript is now acceptable for publication, you may indicate that here to bypass the “Comments to the Author” section, enter your conflict of interest statement in the “Confidential to Editor” section, and submit your "Accept" recommendation.

Reviewer #1: All comments have been addressed

2. Is the manuscript technically sound, and do the data support the conclusions?

Reviewer #1: Yes

3. Has the statistical analysis been performed appropriately and rigorously? 

Reviewer #1: Yes

4. Have the authors made all data underlying the findings in their manuscript fully available?

Reviewer #1: Yes

5. Is the manuscript presented in an intelligible fashion and written in standard English?

Reviewer #1: Yes

6. Review Comments to the Author

Reviewer #1: I have no furthers comments. Thank you for your response. xxxxxxxxxxxxxxxxxxxxxxxxxxxxxxxxxxxxxxxxxx

7. PLOS authors have the option to publish the peer review history of their article (what does this mean?). If published, this will include your full peer review and any attached files.

Reviewer #1: No

---

## [Editor Report · Acceptance letter]

25 Mar 2021

PONE-D-21-00869R1 

Validation of a Polish version of the National Institutes of Health Stroke Scale: Do moderate psychometric properties affect its clinical utility? 

Dear Dr. Wiśniewski:

I'm pleased to inform you that your manuscript has been deemed suitable for publication in PLOS ONE. Congratulations! Your manuscript is now with our production department. 

Kind regards, 

on behalf of

Dr. Miguel A. Barboza 

Academic Editor

PLOS ONE